



# Continental-scale contributions to the global CFC-11 emission increase between 2012 and 2017

Lei Hu[1,2], Stephen A. Montzka[2], Fred Moore[1,2], Eric Hintsa[1,2], Geoff Dutton[1,2], M. Carolina Siso[1,2], Kirk Thoning[2], Robert W. Portmann[3], Kathryn McKain[1,2], Colm Sweeney[2], Isaac Vimont[1,2], David Nance[1,2], Bradley Hall[2], Steven Wofsy[4]

[1] Cooperative Institute for Research in Environmental Sciences, University of Colorado-Boulder, Boulder, CO, USA
[2] Global Monitoring Laboratory, NOAA, Boulder, CO, USA
[3] Chemical Science Laboratory, NOAA, Boulder, CO, USA
[4] Department of Earth and Planetary Sciences, Harvard University, Boston, MA, USA

*Correspondence to*: Lei Hu (lei.hu@noaa.gov)

**Abstract.** The early detection of a global emission increase of CFC-11 after 2012 (Montzka et al., 2018) alerted society to a possible violation of the Montreal Protocol on Substances that Deplete the Ozone Layer (MP). This early alert resulted in parties participating in the MP taking urgent actions (United Nations Environment Programme (UNEP), 2019). As a result, atmospheric measurements made in 2019 suggest a sharp decline in global CFC-11 emissions (Montzka et al., 2021). Despite the success in the early detection and mitigation of some of this problem, regions fully responsible for the recent global emission changes of CFC-11 have not yet been identified. Roughly two thirds (60 ± 40 %) of the emission increase between 2008 - 2012 and 2014 - 2017 and two thirds (60 ± 30 %) of emission decline between 2014 - 2017 and 2019 was explained by regional emission changes in eastern mainland China (Park et al., 2021; Rigby et al., 2019). Here, we used atmospheric CFC-11 measurements made from two global aircraft surveys, the HIAPER Pole-to-Pole Observations (HIPPO) in November 2009 – September 2011 and the Atmospheric Tomography Mission (ATom) in August 2016 – May 2018, in combination with the global CFC-11 measurements made by the U.S. National Oceanic and Atmospheric Administration during these two periods, to derive global and regional emission changes of CFC-11. Our results suggest Asia accounted for the largest fractions of global CFC-11 emissions in both periods, 43 (37 – 52) % during November 2009 – September 2011 and 57 (49 – 62) % during August 2016 – May 2018. Asia was also primarily responsible for the emission increase between these two periods, accounting for 86 (59 – 115) % of the global CFC-11 emission rise between the two periods. Besides eastern mainland China, we find that temperate western Asia and tropical Asia also contributed significantly to global CFC-11 emissions during both periods and likely to the global CFC-11 emission increase between these periods. Besides Asia, the atmospheric observations also provide strong constraints on CFC-11 emissions from North America and Europe, suggesting that each of them accounted for 10 – 15 % of global CFC-11 emissions during the HIPPO period and smaller fractions in the ATom period. For South America, Africa, and Australia, the derived regional emissions had larger dependence on the prior assumptions of emissions and emission changes, due to a lower sensitivity of the observations considered here to emissions from these regions. However, significant increases in CFC-11 emissions from the southern hemispheric lands



were not likely due to the observed increase of north-to-south interhemispheric gradients in
atmospheric CFC-11 mole fractions from 2012 to 2017.
**1. Introduction**
Trichlorofluoromethane, CFC-11, is a potent ozone depleting substance, whose production
has been controlled by the Montreal Protocol since 1987. By 2010, reported global production
and consumption of CFC-11 was near zero (United Nations Environment Programme (UNEP),
2021a, b). Corresponding to the declining production and consumption, global emissions of CFC-
11 declined between 1988 and 2012. By 2012, the global CFC-11 emission magnitude was 50 –
80 Gg yr$^{-1}$ with this range being associated primarily with its uncertain atmospheric lifetime (Engel
et al., 2018). The remaining emissions of CFC-11 were primarily from existing equipment and
insulation foams, known as "CFC-11 banks". However, a large increase of global CFC-11
emission from 2012 – 2017 was discovered (Montzka et al., 2018; Rigby et al., 2019; Montzka et
al., 2021), suggesting illicit CFC-11 production despite the global ban on production and
consumption under the MP beginning in 2010. This surprisingly large increase of CFC-11
emissions attracted great attention from scientists, policy makers, industrial experts around the
world (Montzka et al., 2018; Rigby et al., 2019; Dhomse et al., 2019; Ray et al., 2020; Adcock et
al., 2020; Keeble et al., 2020; Chen et al., 2020), who sought information to enable rapid mitigation
of the unexpectedly enhanced CFC-11 emissions and ensure no significant delay in the recovery
of stratospheric ozone. Despite the international effort to understand the origin of this large global
emission increase of CFC-11, only a portion of the emission rise (60 ± 40 %) could be explained
by emission increases from eastern mainland China (Rigby et al., 2019; Adcock et al., 2020; Park
et al., 2021). It remains unclear where the rest of the global CFC-11 emission increase originated.
Following the initial studies and announcements of anomalous CFC-11 emission increases,
a surprisingly sharp decline in global CFC-11 emissions occurred from 2018 to 2019 (Montzka et
al., 2021). This decline immediately followed the global emission rise and had a similar magnitude
as the emission rise between 2012 and 2017, resulting in the global CFC-11 emission dropping to
the mean 2008 – 2012 value (Montzka et al., 2021). Interestingly, roughly the same proportion of
this emission decrease (60 ± 30 %) can be explained by an emission drop in eastern mainland
China (Park et al., 2021) during this period, as the contribution of eastern mainland China to the
global CFC-11 emission rise earlier (60 ± 40 %).
In this study, we analyzed global CFC-11 measurements made from the HIAPER Pole-to-
Pole Observations (HIPPO) in November 2009 – September 2011, the Atmospheric Tomography
Mission (ATom) in August 2016 – May 2018 (Wofsy, 2018; Bourgeois et al., 2020) and concurrent
CFC-11 measurements from the NOAA's global atmospheric sampling network (Montzka et al.,
2018) and combined them with Lagrangian-based inverse modeling techniques (Hu et al., 2017)
to quantify continental- and regional- scale CFC-11 emission estimates between both periods.
Coincidentally, the timing of the HIPPO and ATom campaigns covered the periods when the
global CFC-11 emissions were at the minimum and maximum before the CFC-11 emission decline
in 2018 – 2019. Hereafter, we will refer November 2009 – September 2011 as the HIPPO period
and August 2016 – May 2018 as the ATom period. Here we further investigate regional
contributions to the global CFC-11 emission rise between these two periods.
**2. Methods**
**2.1. Overview**



To infer regional CFC-11 emissions from observed atmospheric mole fractions, we used a
Bayesian inverse modeling framework following the method described in our previous studies (Hu
et al., 2015; Hu et al., 2017; Hu et al., 2016). In brief, the inverse modeling method assumes a
linear relationship between atmospheric mole fraction enhancements and upwind emissions. The
linear operator is footprints or the sensitivities of atmospheric mole fraction changes to upwind
emission regions that were computed by time-inverted Lagrangian particle models described in
Stein et al. (2015) and Nehrkorn et al. (2010). Because the inverse problem we generally solve is
not fully constrained by the available number of atmospheric observations, the solution in a
Bayesian inversion (Rodgers, 2000) requires initial assumptions about the magnitudes and
distributions of emissions or prior emissions. By assuming that errors between the "true" and prior
emissions and errors between atmospheric mole fraction observations and simulated mole fractions
(using the computed footprints) follow Gaussian distributions, we construct a cost function ($L$)
(Eq. 1) based on the Bayes' Theorem:
$$L = \frac{1}{2}(z - Hs)^T R^{-1}(z - Hs) + \frac{1}{2}\left(s - s_p\right)^T Q^{-1}\left(s - s_p\right) \tag{1}$$
where, $z$ represents the observed atmospheric enhancement relative to the upwind background
atmosphere. $s_p$ and $s$ represent the prior and posterior CFC-11 emissions. $H$ represents the Jacobian
matrix or the first-order partial derivatives of $z$ to $s$. $R$ and $Q$ stand for the model-data mismatch
covariance and prior flux error covariance. The values given to R and Q determine the relative
weight between the prior emission assumptions and atmospheric observations in the final solution.
Here, we used the maximum likelihood estimation method (Hu et al., 2015; Michalak et al., 2005)
and atmospheric observations to directly solve for site-dependent model-data mismatch errors and
prior flux errors. For the aircraft campaigns (HIPPO and ATom), we derive separate model-data
mismatch errors, one for each campaign.

**2.2. Inversions for the HIPPO and ATom periods**
In this section, we describe the detailed observation selection, background mole fraction
estimation that was pre-subtracted from atmospheric observations before inversions, and prior
emission assumptions for the global inversion we conducted for the HIPPO period and the ATom
period using a Lagrangian inverse modeling approach.

**2.2.1. CFC-11 measurements and data selection for global inversion analyses**
All the CFC-11 measurements considered in our global inversion were made by the Global
Monitoring Laboratory, NOAA, through four different sampling and measurement programs: the
global aircraft surveys (flask samples collected during HIPPO and ATom), a global weekly surface
flask sampling program, a global in-situ sampling program, and a biweekly to monthly aircraft
profiling sampling program focused primarily in North America (Fig. 1). CFC-11 measurements
for the ATom campaigns were primarily made by a gas chromatography and mass spectrometry
(GCMS) instrument (named "M3") that was also dedicated for flask-air measurements in the
global weekly surface flask program. Flask-air samples collected from the biweekly to monthly
aircraft profiling sampling program and from the HIPPO campaign were analyzed by another
dedicated GCMS instrument called "M2" and later upgraded to "PR1" in Sep 2014. Hourly in situ
CFC-11 measurements were made by in situ gas chromatography with electron capture detector
instruments (GC-ECDs) located at individual observatories (the Chromatograph for Atmospheric
Trace Species, CATS). All the NOAA CFC-11 measurements were referenced to the same
calibration scale (NOAA-2016) and suite of primary gravimetric standards. However, small



differences were observed in measurements made from the same air samples that were analyzed
by two different instruments (i.e., median differences: 0.7% between M3 and M2 during the
HIPPO period and 0.9% between M3 and PR1 during ATom period; Fig. S1) or measurements
made for samples collected within ±2 hours that were analyzed by M3 flask-air measurements
and CATS in situ measurements, i.e., median differences < 0.2% during the HIPPO and ATom
periods at three relevant sites (Fig. S1). To minimize the influence of these artificial differences
on derived fluxes, particularly because the atmospheric CFC-11 signals associated with changing
emissions were extremely small (Montzka et al., 2021; Montzka et al., 2018), results from M2 and
PR1 were scaled to those from M3. Scaling factors were calculated over 3-month intervals for M2
and PR1 to make them consistent for the same air-sample analyses. For the CATS measurements,
fewer comparison points were available, so scale adjustments to M3 were based on one scaling
factor per site for the HIPPO period and the ATom period, considered separately and derived from
the collocated measurements collected within ±2 hours.
For measurements made during the HIPPO and ATom campaigns, we only include
measurements below 8 km in the global inversions to minimize the influence of stratospheric loss
on measured mole fractions and because high altitude samples typically have less emission
information. Some samples obtained below 8 km still retained a notable stratospheric loss signal,
and these data were also removed from further considerations on the basis of reduced mole
fractions observed for $N_2O$, which is useful for tracing stratospheric influence in an air parcel
owing to its small atmospheric variability and high-precision measurements.
For data obtained in our regular flask-air sampling programs, the inversions included
results from sites that are relatively away from recent emissions, in order to capture emissions
from broad regions. These observations include the weekly surface flask sampling at remote,
globally-distributed locations (Fig. 1) and aircraft profiling in Cook Islands and Alaska, US, and
above 1 km (above ground) over the contiguous US (Fig. 1). Most of our aircraft profiling
sampling was below 8 km above sea level.
To reduce the extremely large computing cost of footprint calculations for surface in situ
sampling, we chose a subset of in situ samples for inversion analyses. We randomly selected one
sample per day from sites such as Barrow, Alaska, US (BRW) and Tutuila, American Samoa
(SMO), and one daytime sample and one nighttime sample each day at Mauna Loa Observatory,
Hawaii, US (MLO). In situ measurements made at Summit, Greenland (SUM) were excluded due
to poor precision of CFC-11 measurements made at this station.
Although many of the observations we used were from remote Pacific and Atlantic Oceans
locations, or from the free troposphere over North America, they did contain above-zero sensitivity
to emissive signals transported from all the continents, as shown in their footprints (Fig. 1); but
the overall sensitivity to emissions from South America, southern Africa, and Australia is low
relative to North American, Europe, and Asia (Fig. 1). Thus, observational constraints on
emissions from North America, Europe, and Asia are stronger and have reduced dependence on
prior assumptions compared to those from South American, Africa, and Australia.
**2.2.2. Footprint simulations**
We used the Hybrid Single-Particle Lagrangian Integrated Trajectory (HYSPLIT) model
driven by the global data assimilation system at a 0.5° resolution (GDAS0.5°), to simulate
footprints for our global inversion analyses. To determine an adequate number of particles needed
for this global simulation, we tested running HYSPLIT backward for 45 days using 5000 and
10000 particles for a subset of observations obtained from the second campaign during ATom



(ATom II). We compared the footprints from these two independent simulations, which are only
different by < 0.05% in the total summed sensitivities. Footprint distributions and magnitudes in
individual time steps are also almost identical, suggesting using 5000 particles was adequate for
our global simulation.
To determine an adequate time duration for each HYSPLIT simulation, we compared
footprints for observations with enhanced CFC-11 mole fractions versus those with relatively low
mole fractions for observations made at different altitudes and latitudes from ATom II. Our results
show that, for observations in all altitudes and latitudes bins, those with enhanced CFC-11 mole
fractions always had higher sensitivity to upwind populated regions in the first 20 days (Fig. S2);
after that, the overall sensitivity was relatively small and constant, likely due to evenly distributed
particles throughout the troposphere beyond 20 days. This result suggests running HYSPLIT for
more than 20 days was likely sufficient for capturing the major emission influence on atmospheric
CFC-11 mole fraction observations made over the remote atmosphere. In the analysis presented
here, sensitivities were derived with HYSPLIT-GDAS0.5° by tracking 5000 particles back in time
for 30 days.
**2.2.3. Estimation of background mole fractions**
Emissions are derived from the Lagrangian analysis from measured mole fraction
enhancements above background values. For each observation, the background mole fraction was
estimated based on the 5000 HYSPLIT-GDAS0.5° back-trajectories and a background mole
fraction field. We derived several background mole fraction fields for comparison. Initially, we
examined the modeled 4D monthly CFC-11 mole fractions from the Whole Atmosphere
Community Climate Model (WACCM) (Davis et al., 2020; Marsh et al., 2013; Montzka et al.,
2021; Ray et al., 2020). In the WACCM simulation, global mole fractions and distributions of
CFC-11 were initialized in the year 1980. The WACCM model was run using version 1.2.2 with
interactive chemistry in the specified dynamics configuration at a resolution of 1.9° latitude × 2.5°
longitude horizontal with 88 vertical levels from Earth's surface to $6 \times 10^{-6}$ hPa. Horizontal winds
and temperatures were nudged to specified dynamics derived from the Modern Era Retrospective-
analysis for Research and Applications (MERRA2). Compared to the ATom observations, these
WACCM forward simulations show 2 ppt positive average biases (Fig. S3), with larger biases in
the higher northern latitudes (~ 4 ppt). We also tried scaling this forward modelled background to
reduce its latitudinal bias. For this purpose, we calculated monthly, latitude-dependent scaling
factors every 30° in latitudes based on the ratios between the WACCM modeled average CFC-11
mole fractions below 3 km and the surface CFC-11 mole fractions observed by the NOAA's long-
term global weekly surface flask-air sampling network. This scaling allowed a reduction of
latitudinal biases in the WACCM simulations (Fig. S3), although a ~2 ppt bias still exists in the
equatorial region. As an alternative, we constructed an empirical 4D CFC-11 mole fraction field
based on NOAA observations. A 4D measurement-based background field was constructed by
propagating a subset of measured mole fractions of CFC-11 from the NOAA's global surface and
ongoing airborne flask-air sampling programs back in time along the 5000 back-trajectories for a
certain time duration. This subset of observations was selected given their measured mole fractions
were lower than a certain threshold in each 30° in latitude x 3 km in altitude box. We tested a
range of thresholds between the 40th and the 80th percentile in the HIPPO and ATom periods. We
ended up selecting observations lower than the 70 – 80th percentile in each box as background
observations during the HIPPO period and ones lower than 40 – 50th percentile as background



observations during the ATom period, so that the inversely derived global emissions in both
periods were consistent with those derived from the global 3 box model with different choices of
atmospheric lifetimes (Montzka et al., 2021). Although the inversely derived global emission
magnitudes were sensitive to the choice of the background threshold, the relative regional emission
distribution or the fraction of regional emissions to the global emission was not. By propagating
this subset of observations back in time, it provided a 4D field of CFC-11 mole fractions that we
then averaged every 5° latitude × 20° longitude × 2 km longitude every half or one month. We
tested the time duration of 10 – 30 days for propagating particles along the 5000 back-trajectories
and half or one month averaging time windows for estimating this empirical background. We
found propagating particles back in time for 10 days with an averaging window of one month can
produce a background field that best agrees with independent ATom and HIPPO observations
(Figs. S3 and S4). We further added the influence of stratospheric air on high altitude CFC-11
mole fractions (8 - 10 km) in the polar regions (> 60°N or > 60°S) to this empirical background,
by considering the vertical gradients simulated by WACCM over these regions. As a result, this
approach produced a background that best agreed with the ATom and HIPPO observations
compared to the previous two approaches (Figs. S3 and S4). Therefore, we used the 4D
background field derived from a subset of CFC-11 measurements in our final global inversion
simulations. Note that, the HIPPO and ATom campaign data were intentionally excluded and used
as independent assessment of the 4D background estimates at first; but they were later included in
the final 4D empirical background construction, after we confirmed that the empirical approach
produced the best estimate of background.
From the 4D measurement-based background mole fraction field, we sampled 5000 mole
fraction estimates at the locations of the 5000 back-trajectories at the end of the 30 days and then
averaged these 5000 mole fraction estimates to obtain one background mole fraction for each
observation. We examined the particle locations at the end of the 30 days using observations
collected at 0 - 8 km from ATom II. For the majority of these observations, 80% - 100 % of
particles were located between 0 and 10 km at the end of the 30 days in the HYSPLIT back-
trajectory runs. For particles that exited from the top at 10 km before 30 days, we sampled the
mole fractions at 10 km when they exited the background mole fraction field.
**2.2.4. Prior emissions**
We constructed 11 different prior emissions for inversion analyses in both HIPPO and
ATom periods (Fig. 2). The first prior, "population_67", was constructed with a global CFC-11
emission of 67 Gg yr$^{-1}$ and the posterior emissions derived for the contiguous US (CONUS) from
Hu et al. (2017). Over the CONUS, the posterior annual 1° x 1° emissions derived for 2014 were
applied to all months in either HIPPO or ATom periods. We then subtracted the annual total
CONUS emissions (~ 4 Gg yr$^{-1}$) from the global total emission of 67 Gg yr$^{-1}$ and distributed the
remaining emissions around the globe based on the Gridded Population of the World (GPW) v4
(https://sedac.ciesin.columbia.edu/data/collection/gpw-v4). The second prior, "population_40",
was the emission distribution in "population_67" with emission magnitudes reduced by 40%
across the globe. The priors "population_87_NA", "population_87_SA", "population_87_Af",
"population_87_Eu", "population_87_Au", "population_87_BA", "population_87_TEA",
"population_87_TWA", and "population_87_TA" incorporated the "population_67" prior and an
additional 20 Gg yr$^{-1}$ of emission imposed over North America, South America, Africa, Europe,
Australia, boreal Asia, temperate eastern Asia, temperate western Asia, and tropical Asia,



respectively. The 20 Gg yr$^{-1}$ of emissions was added to those regions by a constant emission rate
across all the grid cells where emissions were non-zero in the prior of "population_67". The
regions specified as North America, South America, Africa, Europe, Australia, boreal Asia,
temperate eastern Asia, temperate western Asia, and tropical Asia are shown in Fig. 3.
**2.2.5. Inversion ensembles**
We constructed 23 inversion ensembles for deriving global and regional emissions in the
HIPPO and ATom periods. These 23 inversion ensembles included 20 different prior emission
change scenarios between the HIPPO and ATom periods, two background CFC-11 mole fraction
fields, and two sets of observations ("flask only" and "flask + in situ") (Table S1). The 20 prior
emission change scenarios assumed: (scenario 1) no increase of global CFC-11 emissions between
the HIPPO and ATom periods (inversion ensemble IDs #1 - #5 in Table S1); (scenario 2) a 20 Gg
yr$^{-1}$ increase of CFC-11 emissions between the HIPPO and ATom periods, with the increase being
restricted to one of the following regions, respectively: North America, South America, Africa,
Europe, Australia, boreal Asia, temperate eastern Asia, temperate western Asia, and tropical Asia
(inversion ensemble IDs #6 - #14 in Table S1); and (scenario 3) a 20 Gg yr$^{-1}$ decrease of CFC-11
emissions between the HIPPO and ATom periods, with the decrease being restricted to one of the
following regions, respectively: North America, South America, Africa, Europe, Australia, boreal
Asia, temperate eastern Asia, temperate western Asia, and tropical Asia (inversion ensemble IDs
#15 - #23 in Table S1).
In our global inversions, we solved for monthly 1° x 1° emissions and their posterior
covariances at 1° x 1° resolution. Because the uncertainty associated with the 1° x 1° emissions is
large, we aggregated emissions and their posterior covariances into large regional, continental, and
global scales for the HIPPO and ATom periods, considering the cross correlation in errors among
grid cells and across times for each batch inversion (Hu et al., 2017).
**3. Results and Discussion**
**3.1. Increase of CFC-11 emissions between the HIPPO and ATom periods observed in remote**
**atmospheric observations**
The global increase of CFC-11 emissions between 2012 and 2017 was previously derived
from the slow-down in the decline of atmospheric CFC-11 mole fractions observed at Earth's
surface (Montzka et al., 2021; Montzka et al., 2018) and is also shown in Fig. 4 here. Besides at
Earth's surface, a similar magnitude of this slow-down in atmospheric CFC-11 mole fraction
decline is also apparent throughout the free troposphere in the aircraft profiles obtained during the
HIPPO and ATom campaigns (Fig. 4). Here, we calculated the CFC-11 growth rates averaged in
each 30° in latitude × 2 km in altitude box during HIPPO campaigns and during ATom campaigns
separately for samples collected above the Pacific Ocean basin. During HIPPO, we calculated the
average mole fraction differences in each 30° in latitude × 3 km in altitude box between HIPPO
III (3/2010 – 4/2010) and HIPPO IV (6/2011 – 7/2011) and normalized by their time interval to
obtain annual growth rates, whereas we calculated annual growth rates during ATom using the
ATom I (7/2016 – 8/2016) and ATom IV (4/2018 – 5/2018) data. The reason to choose HIPPO
III, HIPPO IV, ATom I, and ATom IV for this calculation is to ensure annual growth rates were
calculated from data collected in similar seasons, so that the impact of seasonal variations in
atmospheric CFC-11 mole fractions on the calculated annual growth rates was similar in both
periods (Fig. S5). Results suggest a median growth rate of -2.5 ppt yr$^{-1}$ between 60°S and 90°N in



the troposphere during the HIPPO period and a median growth rate of -0.7 ppt yr$^{-1}$ during the
ATom period (Fig. 4), indicating a significant increase of CFC-11 growth rates in the troposphere
between the HIPPO and ATom periods.  The impact of atmospheric CFC-11 seasonal cycle
measured at the surface on the calculated changes of annual growth rates between both periods is
about $\pm$0.1 ppt.  Besides the seasonal cycle of atmospheric CFC-11 mole fractions, the Quasi-
Biennial Oscillation (QBO) can also influence atmospheric trace gas mole fractions in the
troposphere (Ray et al., 2020) and thus their growth rates; but this influence was smaller than the
increase of the annual growth rates between the HIPPO and ATom periods, as quantified in
Montzka et al. (2021).
Subtracting the background CFC-11 mole fractions from the selected global CFC-11
observations, there were enhancements up to 3 ppt measured in air above the Pacific Ocean basin
by the global HIPPO and ATom aircraft surveys, the global weekly flask sampling, and the
selected daily – to "every other day" in situ sampling (Fig. 5).  Relatively larger enhancements
were more frequently measured during the ATom period than during the HIPPO period (Fig. 5).
However, the average increase in enhancements of the atmospheric CFC-11 mole fractions
measured during ATom were only 0.2 – 0.3 ppt higher than observed during the HIPPO campaign
(Fig. 5).  The 0.2 – 0.3 ppt increase in the atmospheric CFC-11 enhancements were also
independently measured by the global weekly flask sampling, and in situ sampling networks over
the Pacific Ocean basin (Fig. 5).  Results from HIPPO and ATom suggest that increased mole
fraction enhancements over the Pacific Ocean basin existed  primarily between 0 and 60 ºN (Fig.
5), where the lower and middle tropospheric air mainly contains emissive signals from Eurasia,
western North America, and tropical America (Fig. S6).  Furthermore, when comparing air over
the Pacific Ocean versus the Atlantic Ocean sampled by ATom, air above the Pacific Ocean basin
was more enhanced with CFC-11 than air above the Atlantic Ocean basin (Fig. 5), suggesting
regions immediately upwind of the Pacific Ocean may be emitted more CFC-11 than regions
upwind of the Atlantic Ocean (Fig. 1b) during the ATom period.
**3.2. Regional emissions derived from HIPPO and ATom global inversions**
**3.2.1. The base scenarios with only flask-air measurements**
To quantitatively understand what measured atmospheric CFC-11 variability implies for
global and regional CFC-11 emissions, we conducted Bayesian inversions as described above.  We
first only used the flask-air measurements made by the two GCMS instruments.  These
measurements include samples collected during HIPPO and ATom, the global weekly flask-air
sampling program, and the regular aircraft flask-air sampling program located primarily over
North America.  The inversions derived from these flask-air measurements are referred to here as
"flask-only inversions".  In this first base scenario, we used the same prior emission with a global
CFC-11 emission of 67 Gg yr$^{-1}$ ("population_67") for both HIPPO and ATom periods (Table
S1). The global emissions derived from this scenario (67 $\pm$ 7 Gg yr$^{-1}$ and 87 $\pm$ 9  Gg yr$^{-1}$ for the
HIPPO and ATom periods) were based on background estimates that were calibrated against the
global 3-box model results, such that the global CFC-11 emissions derived from the grid-scale
inversions were consistent with those from the global 3-box model with an atmospheric lifetime
of 52 years reported by Montzka et al. (2021).
An inverse analysis of the flask data obtained during the HIPPO and ATom periods suggest
changes in the total magnitude and distribution of CFC-11 emissions from 2010 to
2018. Significant emission increases were derived for Asia, which we estimate was primarily



responsible for the global CFC-11 emission increase from 2010 to 2018. During the HIPPO period
(November 2009 – September 2011), Asia emitted 35 ($\pm$5) Gg yr$^{-1}$ of CFC-11, accounting for
50% of global CFC-11 emissions, whereas Asian annual CFC-11 emissions increased to 51 ($\pm$8)
Gg yr$^{-1}$ during the ATom period in August 2016 – May 2018, equal to 60% of the global CFC-11
emission at that time. The increase of CFC-11 emission from Asia during these two periods, 16
($\pm$10) Gg yr$^{-1}$, accounted for 80 - 90 % of global CFC-11 emission increases (Fig. 6), as derived
from this scenario.

Our inversion results also suggest that the Asian CFC-11 emissions and emission increases
were primarily contributed by the temperate eastern Asia, temperate western Asia, and tropical
Asia in approximately equal amounts (Fig. 6). Correlations in the posterior emissions among these
three Asian subregions ($r^2$) were less than 0.1, suggesting the inversion was able to separate
regional total emissions from these three regions, although the derived analytical uncertainties
associated with emissions at the subregional level are large (Fig. 6).

Emissions derived for North America, South America, Africa, and Europe were 5 – 15 Gg
yr$^{-1}$ for each region in both the HIPPO and ATom periods. Emissions derived for Australia were
less than 1 Gg yr$^{-1}$. Changes of CFC-11 emissions between both periods derived for these
continents from this scenario were smaller than their associated uncertainties.

With "flask-only" observations, we also tested the sensitivity of posterior regional
emissions to the prior emission magnitude. Here, we considered the second "population-density"
prior with a substantially lower global total CFC-11 emission of 40 Gg yr$^{-1}$ for both periods
("population_40") (Table S1). Derived regional emissions from this second scenario were
consistent with results discussed in the first scenario in both the distribution and total magnitude
of posterior emissions.

### 3.2.2. Scenarios with more observations, prior assumptions, and an alternative background

To increase the observational constraints in the global CFC-11 inversion, we then included
additional observations from the in-situ CFC-11 measurements (Fig. 1). The derived posterior
emissions with this expanded observational dataset (and with the same population-based priors
and background estimates) show slightly higher global emissions, especially from tropical Asia,
during the ATom period (Fig. 7). Besides inclusion of additional observations, we also considered
an alternative background estimate (background 2) that was calibrated to the global CFC-11
emission estimates with alternative atmospheric lifetimes (54 and 56 years) (Montzka et al., 2021)
(Table S1). As expected, the derived global and regional emissions were lower with a background
calibrated to a longer atmospheric lifetime. However, the derived regional contributions to the
global CFC-11 emissions and emission changes between the HIPPO and ATom periods were
consistent with results considering a shorter lifetime (Fig. 7).

Results discussed so far were based on prior emissions with zero changes between the
HIPPO and ATom periods for all regions considered. The remaining questions are whether the
derived near-zero emission changes over North America, South America, Africa, Europe, and
Australia were due to the influence from prior assumption (of zero emission changes) or if they
were constrained by the atmospheric observations. Another question is how much the derived
Asian emissions and emission changes are dependent on prior assumptions. To address these
questions, we constructed 14 additional emission priors that assumed 20 Gg yr$^{-1}$ CFC-11 emission
increases between the HIPPO and ATom (7 priors) or 20 Gg yr$^{-1}$ CFC-11 emission decreases
between both periods (another 7 priors; Table S1; Fig. 8). In the first 7 cases, we considered the





same population-based prior with a global CFC-11 emission of 67 Gg yr$^{-1}$ during the HIPPO
period, whereas during the ATom period, we assumed there was an increase of 20 Gg yr$^{-1}$ of CFC-
11 emissions over individual continents (*i.e.*, North America, South America, Africa, Europe,
Australia) or individual Asian subregions (*i.e.*, boreal Asia, temperate eastern Asia, temperate
western Asia, and tropical Asia) (Table S1; Fig. 8). In the latter 7 cases, we considered opposite
scenarios, where we assumed 67 Gg yr$^{-1}$ of emissions during the ATom period and 87 Gg yr$^{-1}$ of
emissions during the HIPPO period, so that emissions over individual continents or individual
Asian subregions had a 20 Gg yr$^{-1}$ decrease between both periods (Fig. 8). Note that, given we've
already know there was a global increase of CFC-11 emissions from 2010 to 2018 (Montzka et al.,
2021; Montzka et al., 2018) and $60 \pm 40$ % of this global increase was from eastern mainland
China (Park et al., 2021; Rigby et al., 2019), many of the assumed 14 prior cases were quite
unrealistic. However, such extreme cases helped for estimating uncertainties that truly reflect the
capability of the selected atmospheric measurements for constraining continental and regional
emissions and their change through time. In all of the 14 extreme cases, regional emissions and
emission changes derived for the northern hemispheric lands, i.e., Asia, North America, Europe,
were consistent (Fig. 8). Derived regional emissions and emission changes for the southern
hemispheric lands, such as South America, Africa, Australia, however, show a strong dependence
on prior assumptions, especially during the ATom period (Fig. 8). The strong dependence of
inversion-derived emissions over the southern hemispheric lands were due to large sampling gaps
and small sensitivity to emissions from these regions (Fig. 1).
Summarizing emissions derived from all 23 inversion ensembles (Table 1; Figs. 6 - 8), our
results suggest the relatively remote observations provide important constraints on regional
emissions from North America, Asia, and Europe, as the derived ranges of posterior emissions
were smaller than the ranges of prior emissions considered for these regions (Figs. 6 - 8). The
only continent that shows a statistically significant increase of CFC-11 emissions is Asia, where
the best estimate of these 23 cases suggests an increase of 24 $(18 - 28)$ Gg yr$^{-1}$ of CFC-11 emissions
(the $2.5^{th} - 97.5^{th}$ percentile range) (Table 1), accounting for 86 $(59 - 115)$ % of the global CFC-
11 emission increases between the HIPPO and ATom periods. All the best estimates from the 23
inversion ensembles suggest CFC-11 emission increases not only from temperate eastern Asia, but
also from temperate western Asia and tropical Asia. However, if we consider the entire range of
uncertainties (the range of best estimates and $2\sigma$ errors from each inversion), the derived emission
increases were statistically insignificant at the subregion level (i.e., temperate eastern Asia,
temperate western Asia, and tropical Asia).
Our results also suggest inverse modeling of the relatively remote observations we
considered here provided only weak constraints on emissions from the southern hemispheric
continents, i.e., South America, Africa, and Australia. Although we cannot eliminate the
possibility of some increase in CFC-11 emissions from these southern hemispheric regions based
on atmospheric inversion analyses alone, they did not account for the majority of the emission
increase. This is because during $2010 - 2018$, when the global CFC-11 emissions increased, so
did the north-to-south mole fraction difference between the hemispheres (Montzka et al., 2021),
which indicates the emission increase occurred predominantly in the northern hemisphere.
**3.2.3. Comparison of regional emission estimates from other top-down analyses**
Our regional emission estimates of CFC-11 from the global atmospheric CFC-11
measurements made far away from the emissive regions are in a broad agreement with those



estimated from atmospheric observations made closely downwind of the emissive regions (Table
2), which had atmospheric CFC-11 enhancements that were one-two orders of magnitude larger
than those used in this inversion analysis (Park et al., 2021; Rigby et al., 2019; Hu et al., 2017;
Fraser et al., 2020). Emissions estimated for eastern mainland China using measurements made
in South Korea were $5 - 13$ Gg yr$^{-1}$ during $2010 - 2011$ and $12 - 20$ Gg yr$^{-1}$ during 2016 - 2017,
considering the full range of estimates from multiple inversion systems with different transport
simulations (Park et al., 2021). CFC-11 emission estimates for eastern China based on
measurements made in Taiwan were $14 - 23$ Gg yr$^{-1}$ during $2014 - 2018$ (Adcock et al., 2020). In
the current analysis, we estimated CFC-11 emissions from temperate eastern Asia were $5 - 16$ Gg
yr$^{-1}$ during Nov 2009 – Sep 2011 and $9 - 22$ Gg yr$^{-1}$ during August 2016 – May 2018, which agree
well with the published analyses over eastern China, although our definition of temperate eastern
Asia is slightly different from the regions defined in Rigby et al. (2019), Adcock et al. (2020) and
Park et al. (2021).
Previously, we estimated the US emissions of CFC-11 between 2008 and 2014 with more
extensive atmospheric measurements made from towers and aircraft sites from all vertical levels
over North America (Hu et al., 2017). In this analysis, we only used a subset of observations (only
aircraft observations above 1 km above ground) and a coarser resolution of transport models in the
global inversion. While the North American CFC-11 emissions derived here are likely not as
accurate; they did agree within uncertainties with our previous US estimates (Table 2).
Furthermore, CFC-11 emissions derived for Australia are also comparable with estimates
reported by Fraser et al. (2020) using measurements made in Australia (Table 2). Both suggest
CFC-11 emissions from Australia were less than 1 Gg yr$^{-1}$ between 2009 and 2018. Contributions
from Australia to global CFC-11 emissions and emission changes were very small.
Besides temperate eastern Asia, North America, and Australia, we also compared our
derived European CFC-11 emissions for Nov 2009 – Sep 2011 with the value reported by Keller
et al. (2011) for western Europe in 2009. Our best estimate for all of Europe was about twice as
large as reported by Keller et al. (2011) for the western Europe, which only accounted for 40% of
the area we considered for all of Europe. If aggregating emissions from only grid cells considered
in Keller et al. (2011), the aggregated total emissions would be similar to the value reported by
Keller et al. (2011), although both studies focused on two different time periods (Table 2).
Other than the regions mentioned above, previous emission estimates for the rest of the
world are quite limited. Only one study quantified CFC-11 emissions from the northern and central
area of India in June 2016, reporting emissions of $\sim 1 - 3$ Gg yr$^{-1}$ (Say et al., 2019). It is hard to
make a fair comparison with our analysis, given its short analysis period and a much smaller area
than our defined temperate western Asian region (Fig. 4). However, there was observational
evidence indicating likely strong regional emissions and a regional emission increase over
temperate western Asia between $2012 - 2017$. This was shown as substantially enhanced CFC-11
mole fractions observed in temperate western Asia for flask measurements made during 2012 –
2018 (Simpson et al., 2019) and the slow-down of atmospheric CFC-11 decline retrieved from
satellite remote sensing measurements (Chen et al, 2020). Furthermore, in situ measurements
made in tropical Asia in 2017 (Lin et al., 2019) also indicate likely strong regional emissions of
CFC-11 over this area.



## 4. Conclusions

We used global atmospheric CFC-11 measurements primarily made over the Pacific and Atlantic Ocean basins and in the free troposphere over North America to quantify changes in continental-scale emissions between November 2009 - September 2011 and August 2016 – May 2018. These two periods covered the times when the global CFC-11 emissions were at their minimum and maximum, respectively, in recent years, at least before the sharp decline noted after 2018 (Montzka et al., 2021). Atmospheric CFC-11 measurements made in the HIPPO and ATom campaigns confirm that the slow-down of atmospheric CFC-11 mole fraction decline between 2009 and 2018 was present throughout the troposphere. The ATom campaign data further display larger atmospheric CFC-11 enhancements in flights, particularly over the Pacific Ocean basin as compared to the Atlantic Ocean basin, suggesting larger emissions in regions immediately upwind of the Pacific Ocean than the Atlantic Ocean.

Inverse modeling of these global atmospheric CFC-11 measurements suggests three Asian regions were primarily responsible for the global CFC-11 emission changes from 2009-11 to 2016-18 in all of the 23 inversion ensembles, including various extreme initial assumptions of regional CFC-11 emission changes ($\pm$ 20 Gg yr$^{-1}$) between both periods. Our results suggest that, during November 2009 – September 2011, Asia emitted 24 (14 – 40) Gg yr$^{-1}$ of CFC-11, accounting for 43 (37 – 52) % of the global emission (Table 1), whereas the Asian CFC-11 emissions increase to 48 (38 – 65) Gg yr$^{-1}$ or 57 (49 - 62) % of the global emission during August 2016 – May 2018 (Table 1). In both periods, substantial CFC-11 emissions were derived for temperate eastern Asia, temperate western Asia, and tropical Asia. Besides eastern mainland China, our results suggest there could be increases of CFC-11 emissions from temperate western Asia and tropical Asia from 2010 to 2018, considering the range of best estimates from the 23 inversion ensembles. In contrast to Asia, other continents accounted for relatively smaller fractions of global CFC-11 emissions in both periods. Although for continents in the Southern Hemisphere, our inversion analyses only provide weak constraints on the CFC-11 emission changes between 2012 and 2018. However, significant increases in CFC-11 emissions from these regions are unlikely, given the observed concurrent increase of the north-to-south difference in CFC-11 surface mole fractions.

## Acknowledgement

This work was funded by the NASA Earth Venture Atmospheric Tomography (ATom) mission (NNX16AL92A). We thank our retired colleagues Dr. Ben Miller for his development of the Perseus (PR1) GCMS instrument and Dr. James Elkins for his leadership and contribution to the HIPPO and ATom flask sampling and measurements. We also thank Dr. Arlyn Andrews, Dr. Ariel Stein, and Dr. Christopher Loughner for suggestions on HYSPLIT simulations.

**Code/Data availability:** NOAA atmospheric observations are available at the NOAA/GML website (https://gml.noaa.gov/hats/). Data collected from ATom were available via https://espo.nasa.gov/atom/content/ATom. Data collected from HIPPO were available via https://www.nsf.gov/news/news_summ.jsp?cntn_id=127003. Inversion-derived continental fluxes were tabulated and described in this paper. All analysis tools and computing code used in this analysis will be available upon reasonable request.

**Author contributions**



LH and SAM designed the analysis; LH conducted inversions and wrote the paper; SAM led the NOAA global flask measurements, HIPPO and ATom GCMS measurements, and provided substantial input on the analyses and edits of this paper; FM and EH collected HIPPO and ATom flask-air samples; GD led the CATS measurements and prepared CATS data for this analysis; MCS made the NOAA flask measurements; LH and KT computed HYSPLIT footprints; RWP conducted the WACCM simulations and provided the model results; KM conducted NOAA aircraft data QA/QC; CW led the NOAA aircraft sampling network; IV led the Persus GCMS flask measurements; DN helped with data QA/QC for CFC-11 flask measurements; BH led the calibration for NOAA measurements; SW led the HIPPO and ATom campaigns; all authors contributed to the editing of this paper.

**Competing interests:** the authors declare no competing interests.

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





**Table 1.** Global and regional emissions (Gg yr$^{-1}$) derived from this analysis for Nov 2009 – Sep
2011 and Aug 2016 – May 2018 and the derived emission increases between the two periods (left
columns). Two types of uncertainties were given in the parentheses. The former range indicates
the 2.5$^{th}$ – 97.5$^{th}$ percentile range of the best estimates derived from the 23 inversion ensembles.
The latter range indicates the 2.5$^{th}$ – 97.5$^{th}$ percentile range of the best estimates with their 2$\sigma$
analytical uncertainties from each inversion ensemble member. The right columns indicate the
fractional contributions of regional emission to the global CFC-11 emissions and emission
changes. Values in the parentheses for the right columns indicate the 2.5$^{th}$ – 97.5$^{th}$ percentile range
from the 23 inversion ensembles. Definitions of various regions are shown in Fig. 3.

| Region | Nov 2009 - Sep 2011 | | Aug 2016 - May 2018 | | Change | |
|---|---|---|---|---|---|---|
| | Emissions | Percentage | Emissions | Percentage | Emissions | Percentage |
| **Global** | 56 (49 – 68; 39 - 75) | 100 | 84 (78 – 101; 67 – 113) | 100 | 29 (21 – 40; 5 – 56) | 100 |
| *Continents* | | | | | | |
| N. America | 5.9 (5.6 – 7.1; 4.4 - 8.5) | 11 (9 - 14) | 5.6 (5.1 – 7.5; 3.5 – 9.6) | 7 (6 - 9) | -0.4 (-2 – 1; -4 - 4) | -1 (-5 - 5) |
| S. America | 6 (5 – 10; 1 - 16) | 11 (9 - 16) | 9 (7 – 18; 3 - 25) | 11 (8 - 18) | 3 (-2 – 11;-9 – 19) | 8 (-9 - 27) |
| Africa | 10 (7 – 14; 1 - 23) | 17 (13 - 24) | 9 (7 – 14; 2 - 24) | 11 (8 - 15) | -1 (-6 – 5; -17 – 15) | -3 (-26 - 14) |
| Asia | 24 (21 – 33; 14 - 40) | 43 (37 - 52) | 48 (45 – 56; 38 - 65) | 57 (49 - 62) | 24 (18 – 28; 8 - 39) | 86 (59 - 115) |
| Europe | 9 (5 – 11; 2 - 15) | 15 (11 - 20) | 11 (7 – 15; 4 - 18) | 12 (9 - 16) | 2 (-2 – 5; -7 - 10) | 7 (-7 - 19) |
| Australia | 0.5 (0.4 – 2; -1 - 4) | 1 (1-3) | 1 (0.6 – 6; 0.1 - 10) | 1 (1-7) | 0.7 (-1 – 6; -4 - 11) | 2 (-4 - 16) |
| *Asian Subregions* | | | | | | |
| Boreal Asia | 0.6 (0.2 – 3; 0.1 - 5) | 1 (0 - 6) | 0.8 (0.4 – 3; 0.1 - 4) | 1 (0 - 3) | 0.1 (-3 – 2; -4 - 4) | 0 (-11 - 8) |
| Temperate E. Asia | 10 (8 – 13; 5 – 16) | 18 (15 - 21) | 14 (12 – 18; 9 - 22) | 17 (14 - 23) | 4 (2 – 8; -3 - 12) | 15 (6 - 34) |
| Temperate W. Asia | 6 (4 – 10; -3 - 16) | 10 (7 - 14) | 16 (12 – 20; 5 - 29) | 19 (15 - 23) | 10 (6 – 13; -3 - 24) | 36 (25 - 56) |
| Tropical Asia | 8 (6 – 11; 2 - 16) | 14 (11 - 18) | 18 (16 – 23; 11 - 29) | 21 (17 - 25) | 10 (5 – 14; -2 - 22) | 35 (22 - 51) |






**Table 2.** Comparison of regional emissions derived from this study and reported by previous top-
down analyses.

| Regions | Time Periods | Emissions (Gg/y) | References |
|---|---|---|---|
| ***Asia*** | | | |
| Eastern Mainland China | 2008 - 2012 | 5 – 13 [1] | Rigby et al., 2019; Park et al., 2021 |
| Temperate Eastern Asia | Nov 2009 - Sep 2011 | 10 (5 - 16) | This Study |
| Eastern Mainland China | 2014 - 2017 | 12 – 20 [1] | Rigby et al., 2019; Park et al., 2021 |
| Eastern China | 2014 - 2018 | 19 ± 5 | Adcock et al., 2020 |
| Temperate Eastern Asia | Aug 2016 - May 2018 | 14 (9 - 22) | This Study |
| | | | |
| ***Europe*** | | | |
| 35º - 55ºN; -10º - 30ºE | 2009 | 4.2 (2.9 - 5.4) | Keller et al., 2011 |
| 35º - 70ºN; -10º - 60ºE | Nov 2009 - Sep 2011 | 10 (6 - 16) | This Study |
| | | | |
| ***Australia*** | | | |
| Australia | 2010 - 2017 | 0.32 ± 0.04 | Fraser et al., 2021 |
| Australia | Nov 2009 - Sep 2011 | 0.4 (0 - 0.8) | This study |
| Australia | Aug 2016 - May 2018 | 0.6 (0.1 - 1.6) | This study |
| | | | |
| ***North America*** | | | |
| The contiguous US | 2009 - 2011 | 8.2 ± 1.0 | Hu et al., 2017 |
| North America | Nov 2009 - Sep 2011 | 5.9 (4.4 - 8.5) | This study |
| The contiguous US | 2014 | 4.5 ± 0.7 | Hu et al., 2017 |
| North America | Aug 2016 - May 2018 | 5.6 (3.5 - 9.6) | This study |

Notes: [1]values were taken from the reported inversion ensemble spread.



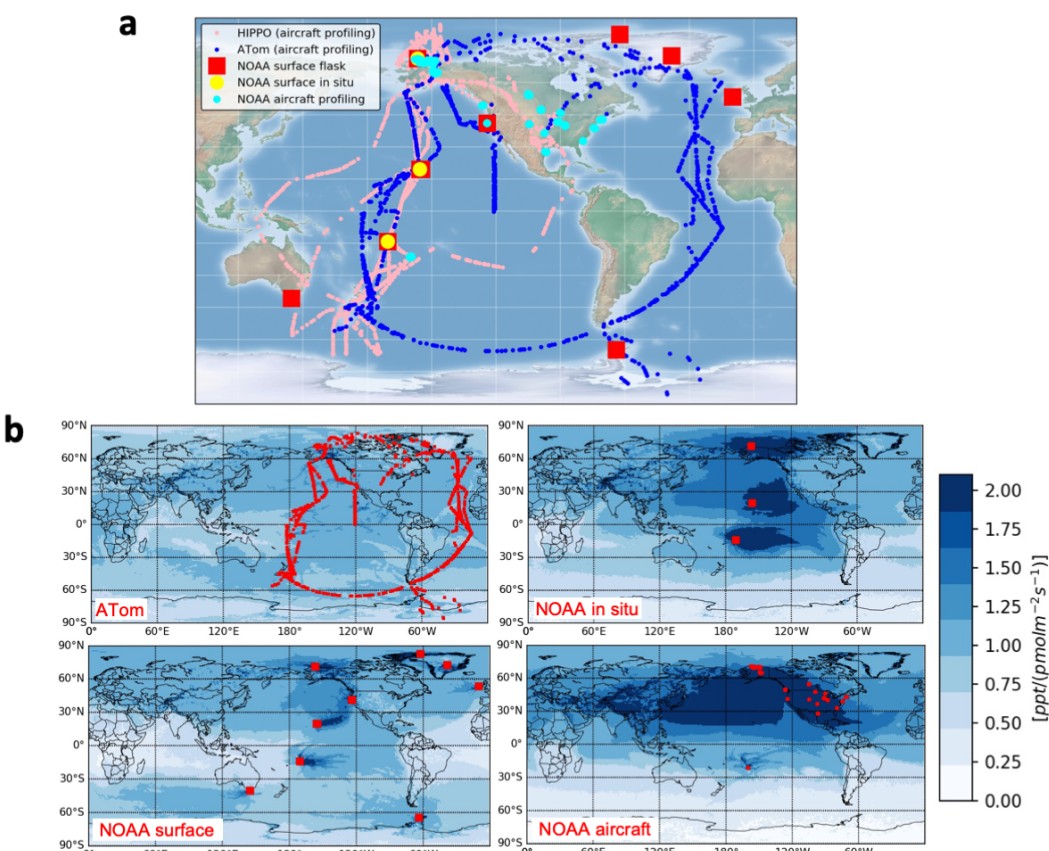

**Fig. 1.** Global atmospheric CFC-11 observations considered in this study, including flask
measurements from the NASA HIPPO and ATom campaigns and the selected observations from
the NOAA global weekly surface flask sampling network, NOAA global in situ surface sampling
network, and NOAA aircraft profiling sites. The bottom panels indicate the summed footprints
between Aug 2016 – May 2018 from ATom (number of observations: 1003), NOAA weekly
surface flask network (number of observations: 781), in situ network (only selected 1 – 2 samples
per day; number of observations: 2559), and biweekly – monthly aircraft profiling sites (only data
above 1 km above ground were selected at North American sites; number of observations: 4824).

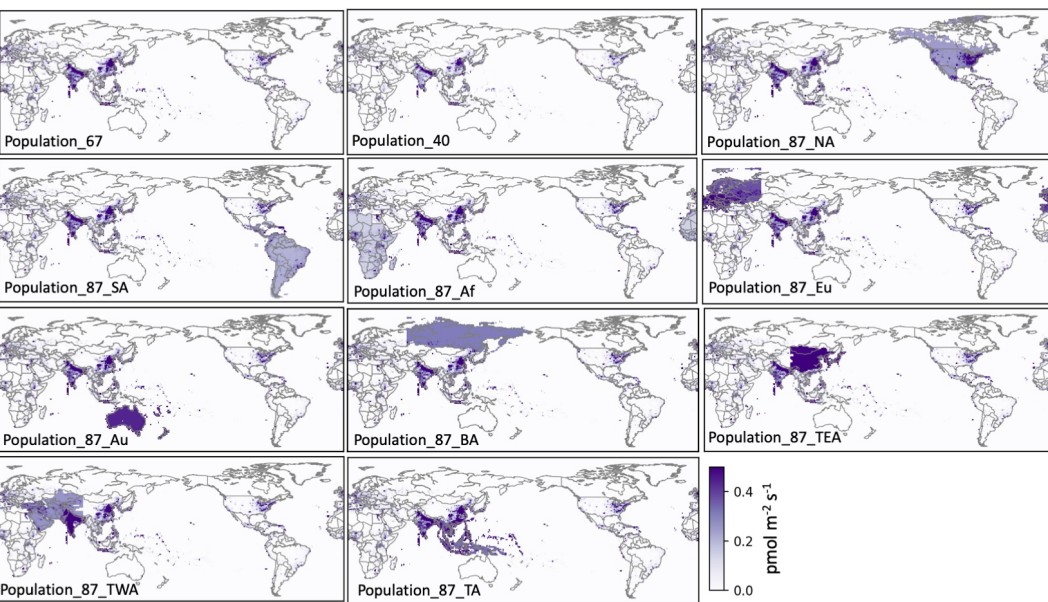

**Fig. 2**. Prior CFC-11 emissions used in this study. Priors of "population_67" and "population_40"
have global CFC-11 emissions of 67 Gg yr$^{-1}$ and 40 Gg yr$^{-1}$. Compared to the prior
"population_67", priors of "population_87_NA", "population_87_SA", "population_87_Af",
"population_87_Eu", "population_87_Au", "population_87_BA", "population_87_TEA",
"population_87_TWA", and "population_87_TA" have a global emission total of 87 Gg yr$^{-1}$ with
additional 20 Gg yr$^{-1}$ emissions imposed over North America, South America, Africa, Europe,
Australia, boreal Asia, temperate eastern Asia, temperate western Asia, and tropical Asia,
respectively.



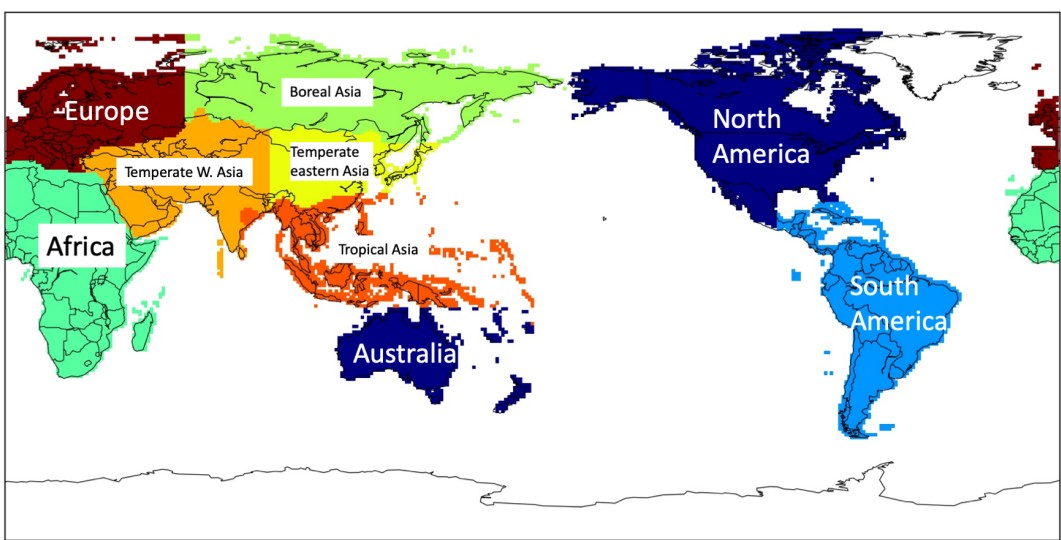

**Fig. 3.** Emissive regions defined for this analysis: North America, South America, Europe, Africa,
Australia, and Asia; Asia was further divided into Boreal Asia, Temperate Eastern Asia, Temperate
Western Asia, and Tropical Asia.



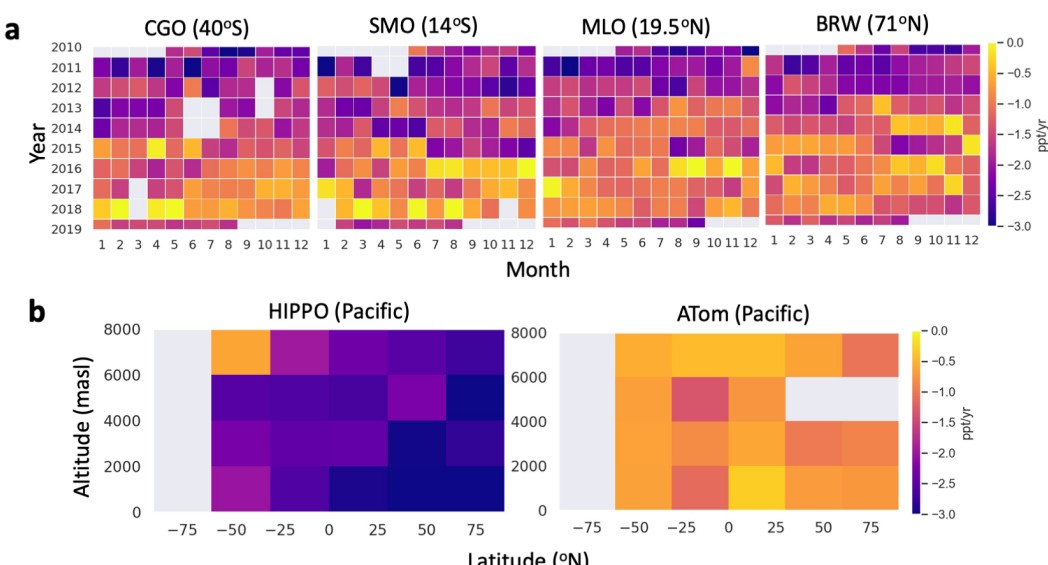

**Fig. 4**. Annual growth rates of atmospheric CFC-11 measured by four surface flask sampling sites over the Pacific Ocean basin from 2010 – 2019 (a) and CFC-11 growth rates measured during the HIPPO and ATom aircraft profiling surveys (b). Each cell indicates an annual difference relative to the prior year for that given month (in panel a) or location (in panel b). Gray cells indicate periods or locations with no data. The four surface sites plotted in panel (a) are at Cape Grim, Tasmania, Australia (CGO), Tutuila, American Samoa (SMO), Mauna Loa, Hawaii, United States (MLO), and Barrow, Alaska, United States (BRW).

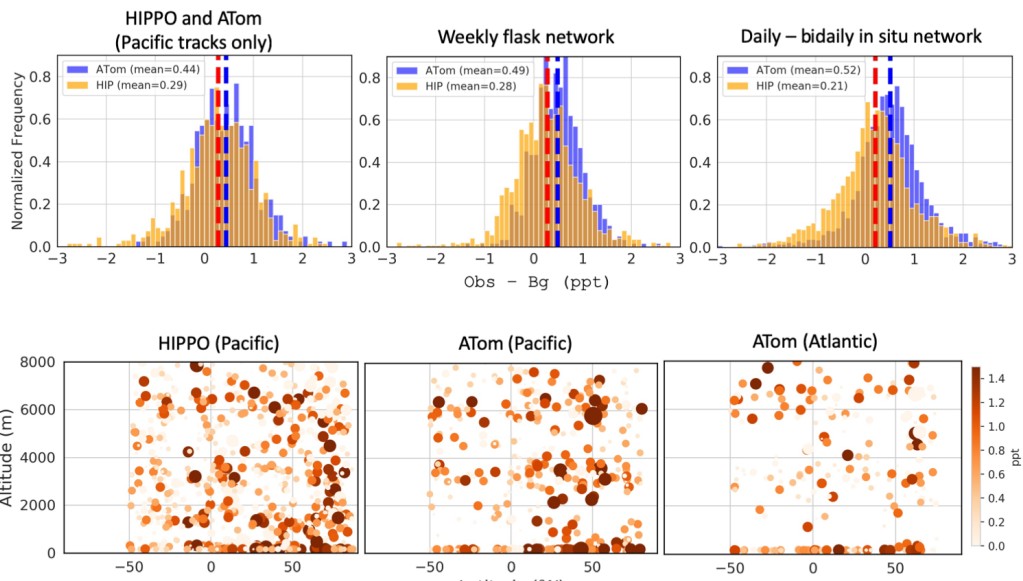

**Fig. 5**. Enhancements of CFC-11 mole fractions relative to background air mole fractions, measured by three independent networks during Nov 2009 – Sep 2011 (HIPPO period) and Aug 2016 – May 2018 (ATom period). (a) Histograms of enhancements of CFC-11 mole fractions measured from flasks collected over the Pacific Ocean basin during the HIPPO and ATom campaigns (left panel), in flasks collected in the NOAA weekly surface sampling network during those periods (middle panel), and measured from the NOAA in situ sampling network in both periods (right panel). Orange bars indicate normalized frequencies of enhancements observed in the HIPPO period, whereas blue bars indicate normalized frequencies of enhancements observed in the ATom period. Red and blue dashed lines denote the mean mole fractions observed during HIPPO and ATom periods. (b) Atmospheric CFC-11 mole fraction enhancements measured from flasks above the Pacific Ocean Basin during HIPPO (left) and ATom (middle), and above the Atlantic Ocean Basin during ATom (right). Both color shading and size of the symbols are proportional to the magnitude of mole fraction enhancements.



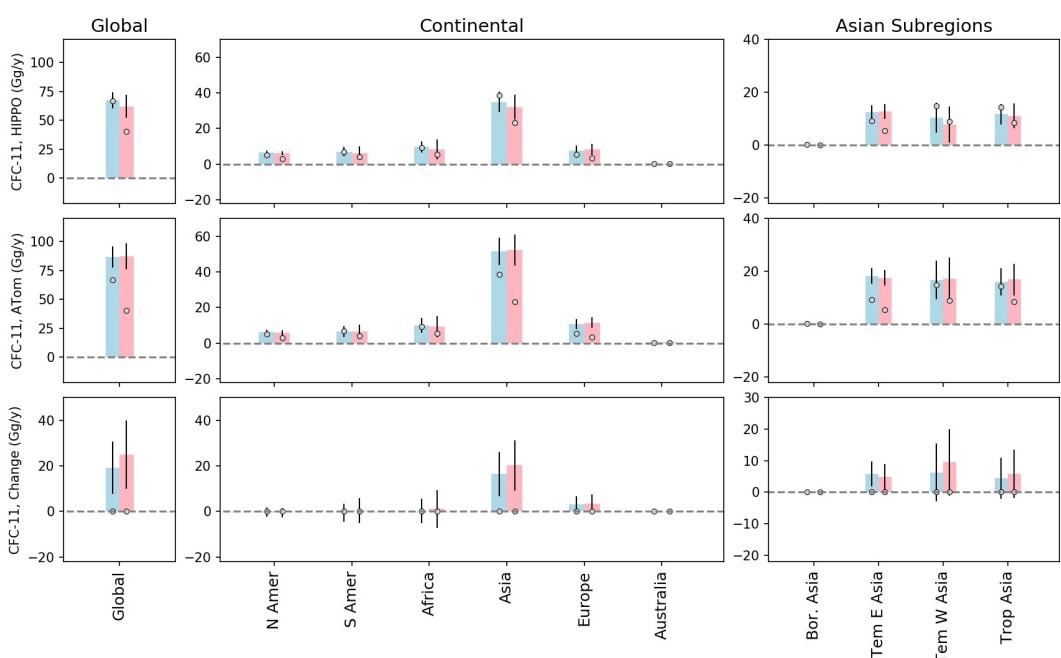

**Fig. 6.** Prior (circles) and posterior (bars) CFC-11 emissions derived for the globe, continents, and Asian subregions, from the "flask-only" inversions for the HIPPO period (upper three panels), the ATom period (middle three panels), and emission differences between the two periods (lower three panels). In each region and from the left to right, open circles denote the two assumed prior emissions ("population_67" and "population 40") with zero changes between the HIPPO and ATom periods; light blue and pink bars correspond to posterior emissions derived from the two different priors. Errorbars of CFC-11 emissions derived for the HIPPO and ATom periods (the upper and middle panels) indicate $2\sigma$ uncertainties derived from individual inversions. Errorbars for the derived CFC-11 emission changes (the lower panels) between the HIPPO and ATom periods were calculated from the sum of $2\sigma$ errors derived for the HIPPO and ATom inversions.

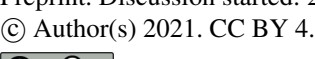



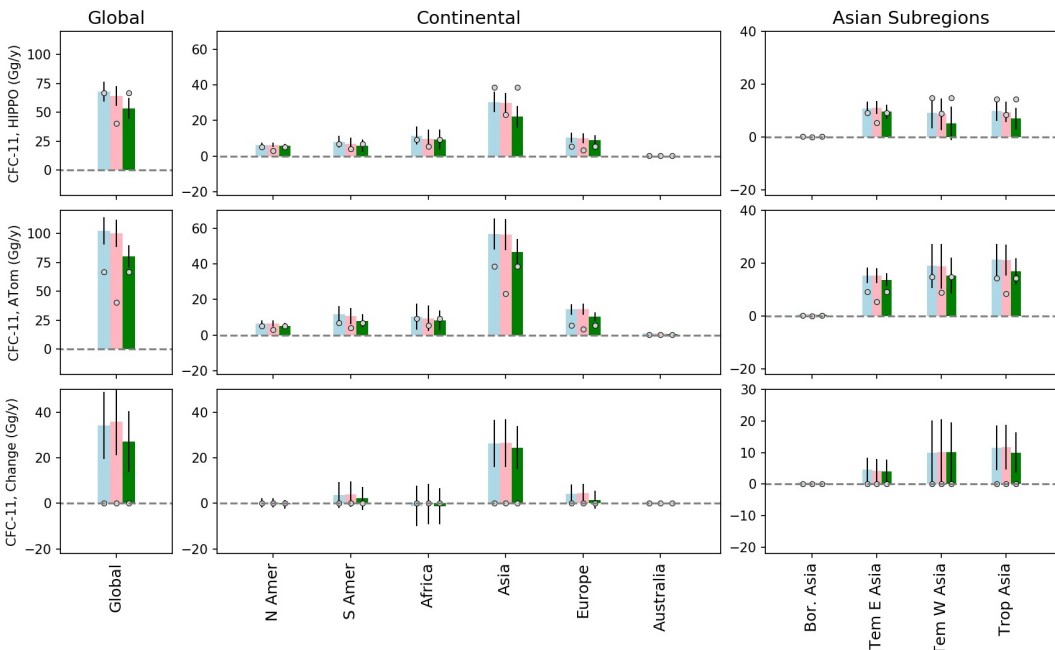

**Fig. 7.** Prior (circles) and posterior (bars) CFC-11 emissions derived for the globe, continents, and
Asian subregions, from the "flask + in situ" inversions for the HIPPO period (upper three panels),
the ATom period (middle three panels), and emission differences between the two periods (lower
three panels). In each region and from the left to right, open circles denote the three assumed prior
emissions ("population_67", "population 40", and "population_67") with zero changes between
the HIPPO and ATom periods; light blue, pink, and dark green bars indicate posterior emissions
derived from the three priors and two different background, as described in inversions ensembles
#3 - #5 in Table S1. Errorbars of CFC-11 emissions derived for the HIPPO and ATom periods
(the upper and middle panels) indicate $2\sigma$ uncertainties derived from individual inversions.
Errorbars for the derived CFC-11 emission changes between the HIPPO and ATom periods were
calculated from the sum of $2\sigma$ errors derived for the HIPPO and ATom inversions.

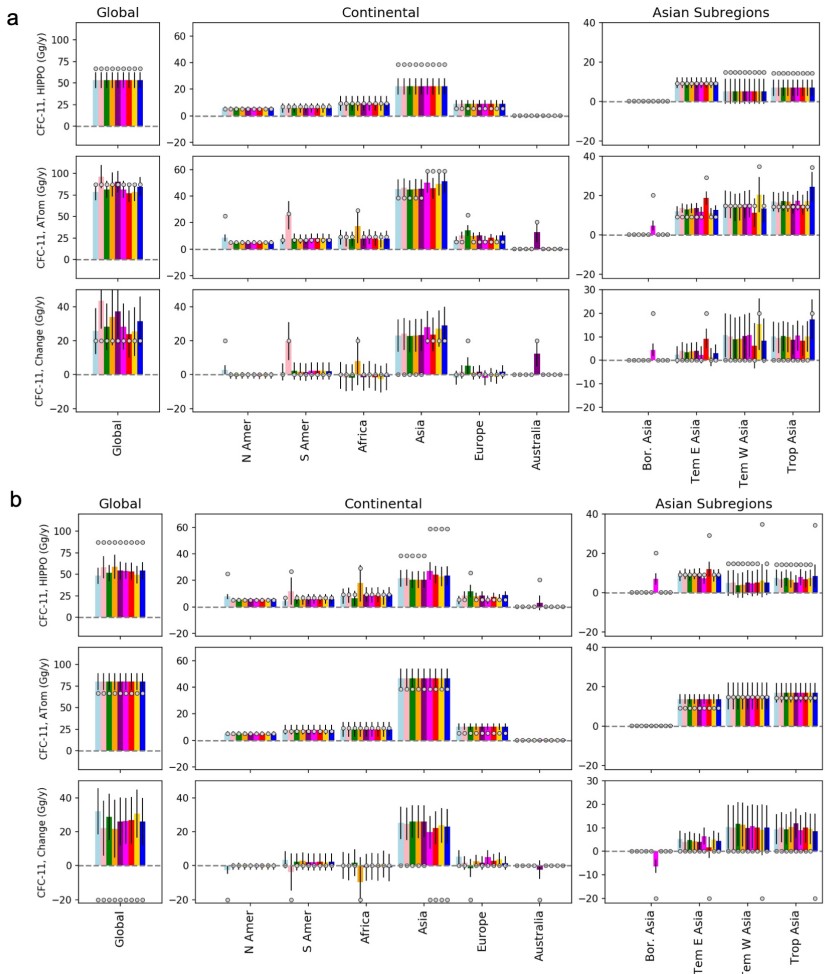

**Fig. 8.** Testing the sensitivity of assumed prior emission changes on the inversion-derived emission changes. (a) Assume a 20 Gg yr$^{-1}$ emission increase between the HIPPO and ATom periods in individual continents and Asian subregions. (b) Assume a 20 Gg yr$^{-1}$ emission decrease between the HIPPO and ATom periods in individual continents and Asian subregions. Similar to Fig. 7, posterior CFC-11 emissions were derived from the "flask + in situ" inversions for the HIPPO and the ATom periods. In each region and from the left to right, open circles denote the prior emissions as described for inversions ensembles #6 - #14 in Table S1 for panel (a), and for inversions ensembles #15 - #23 in Table S1 for panel (b); different color bars indicate the corresponding posterior emissions derived from inversions ensembles #6 - #14 (a) and ensembles #15 - #23 (b). Errorbars of CFC-11 emissions derived for the HIPPO and ATom periods indicate 2$\sigma$ uncertainties derived from individual inversions. Errorbars for the derived CFC-11 emission changes between the HIPPO and ATom periods were calculated from the sum of 2$\sigma$ errors derived for the HIPPO and ATom inversions.