# Peer review of "Continental-scale contributions to the global CFC-11 emission increase between 2012 and 2017"

_Atmospheric Chemistry and Physics, 2021_

## Author Comment (AC1)

**Response to comments from RC1 posted on 8 Oct 2021:**

We appreciate the reviewer for reviewing our manuscript and bringing up his/her invaluable perspectives to this work. We made modifications according to the suggestions. Below describes our line-by-line response for each comment.

Review of the article: "Continental-scale contributions to the global CFC-11 emission increase between 2012 and 2017" by Hu et al.

The manuscript presents inverse modelling results for emissions of CFC-11 at a continental scale in order to better determine the source of a prior-reported global emissions increase. It uses measurements from two aircraft campaigns (for two different periods) and measurements from NOAA's high frequency and flask network under an emsemble of measurement and a priori emission scenarios. This manuscript will be well received due to the mystery of the source of increased CFC-11 emissions, likely due to unreported production. My only major concern is that the prior uncertainties are not clearly communicated, and as such it is impossible to deduce whether the inversion is actually providing any new information. This must be seen before conclusions can be judged. I hope this is just a matter of clarification, and I hope to see the eventual publication of the manuscript.

Response: Thanks for suggesting it. The prior uncertainties and uncertainty reduction are indeed useful metrics to assess how much constraint the observations provide for quantifying regional emissions. We now added information in Section 2.2.4, Fig. 6, and Fig. 7 to describe the prior uncertainties. We also added a short paragraph in Section 3.2.1 and Table S2 to describe the uncertainty reduction between our prior and posterior emission estimates.

Below are a number of suggestions for revisions to improve the manuscript, followed by technical comments, before the paper is suitable for publication.

1. Section 2.2.3: This section can be cut down drastically. Rather than providing information on all approaches tried, simply state the one approach taken in the work. It is useful to hear the alternatives used, but is much better suited to relegation to supporting information.

Response: We reduced this section and only kept the final approach taken.

Section 2.2.4: The use of e.g. "population_87_NA" is confusing as the reader is provided with no information of what that is. They also do not appear anywhere else in the text (perhaps save the definitions in an expansion of the caption for Fig 2). This whole section would be much clearer without the use of the "population_XX" terminology. Further, "prior emissions" are a distribution, not a single value. What you are referring to here is the prior mean(/median/mode as it's Gaussian). 'A priori' may be another useful term if defined. This applied throughout the manuscript. There is also no information on the uncertainty given in your prior distribution, which

makes it impossible to understand the information content/uncertainty reduction provided by the measurements. I suggest providing this information in Table 1 and Fig 6/7.

Response:  We modified this paragraph and used "a priori" where appropriate.  We also added detailed descriptions on prior emission uncertainties and in Figs. 6 and 7.  Because we did use terminology e.g., ""population_87_NA" in our discussion section, it would be useful to keep.  But to avoid the confusion and improve the flow of this paragraph, we removed this term in the middle of Section 2.2.4.  Instead, we explained it at the end of the paragraph.

Lines 366-375: It is stated that $r^2$ is less than 0.1. It's unclear which definition of $r^2$ is used here: there are multiple, some of which can yield negative $r^2$. As such, it's not clear whether there is in fact negative correlation between the regions.

Response:  $r^2$ is a common statistical term that represents correlation between two variables. They range between 0 and 1 and cannot be negative. The reviewer may be confused here with the correlation coefficient, which can be negative.  To avoid further confusion, we made small modifications to improve the clarity.

2.  It is not fully clear to me how the emissions and uncertainties are derived from the ensembles. In Table 1, I assume that the 'range of the best estimate' refers to the range in the MAP solution across the ensemble members? I don't understand the definition of the $2^{nd}$ set of uncertainties – is this the range of the lower/upper 2-sigma values? This needs a clearer explanation in the text. It's not currently clear to me whether the posterior uncertainty for a given ensemble member is greater or smaller than the ranges presented.

Response: In Table 1, the first uncertainty is calculated as the $2.5^{th} – 97.5^{th}$ percentile range of the mean emissions ($\mu_i$) derived from the 23 inversions. This was considered as our "best estimate".  The second uncertainty includes ($2\sigma_i$) derived from the inversions.  The lower bound of the second uncertainty was calculated as the $2.5^{th}$ percentile of $[\mu_1 - 2\sigma_1, \mu_2 - 2\sigma_2, ..., \mu_{23} - 2\sigma_{23}]$ and the upper bound was calculated as the $97.5^{th}$ percentile of $[\mu_1 + 2\sigma_1, \mu_2 + 2\sigma_2, ..., \mu_{23} + 2\sigma_{23}]$.  We also added this clarification in Section 2.2.5 in our revision.

**Technical comments:**

Abstract: ACP guidelines state that there should be no references in the abstract unless urgently required. I do not believe that to be the case here. Acronyms defined in the abstract should be defined again in the main text.

Response: We removed the references in the Abstract.

Line 17: "early detection" is subjective. Many wouldn't consider 8 years later as early detection. Please amend.

Response: We deleted "early" in the abstract.

Line 19: "parties to the MP"

Response: Changes were made as suggested.

Line 60: "policy makers and industrial experts"

Response: Changes were made as suggested.

Line 66: What is the definition of 'eastern mainland China'?

Response: the specific definition of 'eastern mainland China' was described in Rigby et al. (2019) and Park et al. (2021), which are cited here. No change was made regarding this comment.

Line 79: NOAA should be defined

Response: NOAA is now defined.

Line 81: What's the definition of "regional-scale" here?

Response: Regional scale is a scale smaller than a continental scale but larger than state- and city-scales.

Line 86: An outline of the structure of the paper would be useful here.

Response: The structure of the paper can be conveniently reviewed from the headers of the sections. An outline of the structure would be redundant.

Line 91: Were the exact authors included in all studies defined as 'ours'? If not, delete 'our'.

Response: We deleted 'our'.

Line 93: Upwind from the measurement location

Response: We added "upwind from the measurement location".

Line 94: Change "is footprints or" to ", termed footprints, are".

Response: We made changes as suggested.

Line 96: State that these models are HYSPLIT and WRF-STILT.

Response: We actually only used the HYSPLIT model in this study. It is now stated in the manuscript here.

Line 97: Pedantic, but Bayesian inversions *always* require prior probabilities, regardless of the constraints.

Response: We removed "Because the inverse problem we generally solve is not fully constrained" in the revision.

Line 104: Here z is an enhancement. Please reference here the later section on how you derive the enhancement from the measurement?.

Response: We now referenced Section 2.2.3 here.

Line 109: Again pedantic, but "maximum likelihood estimation" is by definition non-Bayesian. If Bayesian, you are finding the maximum a posteriori (MAP) solution.

Response: This seems to be a comment.  No change was made regarding this comment.

Line 130: Be consistent with the use of hyphenation of in situ/in-situ (I would remove it).

Response: We corrected all the uses of "in-situ" to "in situ" throughout the paper.

Line 155: Change 'our' to 'the'

Response: We changed 'our' to 'the' here.

Line 156: relative to what? Best to give an approximate distance for how [far] away the sampling is from emissions.

Response: We now described that "relatively away from recent anthropogenic emissions (e.g., miles away from populated areas or not in the boundary layer)".  We further explained "These observations include the weekly surface flask sampling at remote, globally-distributed locations (Fig. 1) and aircraft profiling in Cook Islands and Alaska, US, and above 1 km (above ground) over the contiguous US (Fig. 1).  Most of our aircraft profiling sampling was below 8 km above sea level. ".

Line 219: How long were they run back in time?

Response: We now added "run back for 10 days" here.

Line 344: Change "described above" to "described in Sect. 2".

Response: We changed it to "described in Section 2" as suggested.

Line 363: It seems incorrect to me that 16 ± 10 Gg/yr then only has a percentage uncertainty of 80-90% of the global total.

Response: In this scenario, the derived global changes are 19 ($\pm$12) Gg/yr (2 sigma). The increase of Asian emission relative to the global increase is 16/19 = 84%. The uncertainty is calculated as (16-10)/(19 – 12) and (16+10)/(19 +12). Because we only kept one significant figure. It is 80- 90% here.

Line 411: Change "given we've already know[n]" to "given it is known".

Response: We changed it to "given it is known".

Line 431: How are you defining the 'best estimates'?

Response: The best estimates were defined as the 2.5[th] – 97.5[th] percentile range of the mean emissions from the 23 inversion ensembles. It was now described in the Method section (2.2.5)

---

## Author Comment (AC2)

**Response to comments from RC2 posted on 21 Nov 2021:**

We appreciate the reviewer for reviewing our manuscript and bringing up his/her invaluable perspectives to this work. We made necessary modifications according to some of the suggestions. Below describes our line-by-line response for each comment.

This study analyzed the atmospheric CFC-11 measurements from two global aircraft surveys - the HIPPO (2009-2011) and Atom (2016-2018) campaigns, to estimate regional scale CFC-11 emissions and the emission changes between two campaign periods. The manuscript demonstrates how a well-designed aircraft measurement can be used to constrain regional emissions estimates.

Overall, the writing and figures are clear, and the methodology maximizes the functionality of high-quality datasets. I encourage the publication of this important work, with only a few minor considerations suggested below.

General comments

1. Global emissions: authors provided their estimates of global CFC-11 emissions for 2009-2011 and 2016-2018 periods in table 1. They were very briefly mentioned in lines 351-355. Authors need to describe more in details how they were determined, which datasets were used for the analysis, and how well consistent they were with the estimates from other studies.

Response: We have used the entire Section 2 to describe how we derived the global and continental emissions from our inversion methods. Section 2.2.1 described the atmospheric datasets that were used for the inversions. Our global estimates agree with other studies (Montzka et al., 2018 and 2021). But they are not completely independent of other estimates because we intentionally chose background thresholds to yield global estimates that are consistent with other studies. Although the inversely derived global emission magnitudes were sensitive to the choice of the background threshold, the relative regional emission distribution or the fraction of regional emissions to the global emission was not. This was all described in Section 2.2.3. No changes were made regarding this comment.

2. Data selection: authors stated that the data included most of the aircraft profiling sampling below 8km. Then does it mean that the HYSPLIT model used to simulate footprint for inversion was also run from the surface boundary layer up to the 8-km altitude? How were the uncertainties associated with the HYSPLIT model analysis analyzed?

Response: In our HYSPLIT configuration, we allowed the particles to travel between 0 – 20 km above sea level. The surface sensitivity was only computed based on the particle density within the planetary boundary layer. The uncertainties associated with HYSPLIT transport simulations were accounted for in the model-data mismatch parameter or the R matrix,

which was calculated from the maximum likelihood estimation using atmospheric observations.

3. Prior emissions: since CFC-11 is an anthropogenic compound, it is reasonable to take population density-based distributions of the global CFC-11 emissions of 67 Gg/yr as prior emissions. But as a base case, authors may need to consider including area-based distributions.

Response: We distributed the global total emission, 67 Gg/yr, based on a global $1^\circ \times 1^\circ$ gridded population density product, which is an area-based product and has a unit of persons per $m^2$. We can not think of another area-based product that is more relevant to this CFC-11 problem.

4. 6-8: it was stated that the error bars for the emissions changes between the HIPPO and ATom periods were calculated from the sum of $2\delta$ errors derived for the HIPPO and ATom inversions. But propagated errors from a subtraction can be determined by the square root of the sum of the squares of each error. So, the errors shown in the lowest panels of Figs. 6-8 might be overestimated.

Response: Thanks for pointing this out. We actually calculated the error as the square root of the sum squares of each error in the upper and middle panels. The caption for Fig. 6 – 8 was incorrect. We fixed the captions now.